# Distinguishing Imitation Error from Intrinsic Motion Learning Difficulty

**Zhaorui Meng** [1]  **Lu Yin** [1]  **Xinrui Chen** [1]  **Chengxu Zuo** [1]  **Anjun Chen** [1]  **Shihui Guo** [1]  **Yipeng Qin** [2]

## Abstract

Physics-based motion imitation is central to humanoid control, yet current evaluation metrics (e.g., MPJPE) only quantify imitation outcomes, not their underlying causes. This conflation obscures a critical diagnostic question: *when imitation error occurs, does it stem from policy limitations or the intrinsic learning difficulty of the target motion?* To resolve this ambiguity, we propose the **Torque Variation Score (TVS)**, a physics-grounded metric that quantifies the inherent learning difficulty of a motion independently of any policy's performance. TVS measures the magnitude of torque variation required to correct small pose perturbations, directly capturing how dynamical properties shape the reinforcement learning landscape. We establish that high-TV motions induce flat reward landscapes and vanishing policy gradients, explaining persistent imitation failures. Extensive experiments with state-of-the-art methods (UHC, PHC+) confirm TVS strongly correlates with imitation error and enables principled error attribution: high error on low-TV motions indicates policy deficiency, while high error on high-TV motions reflects fundamental learning constraints. Beyond error diagnosis, TVS facilitates three practical applications: *Maximum Imitable Difficulty (MID)* for policy capability assessment, *Difficulty-Stratified Joint Error (DSJE)* for granular performance profiling, and *Flawed Motion Detection* for identifying segments with abnormally high learning difficulty to support mocap data curation and quality control. TVS provides a rigorous lens to distinguish policy-induced errors from motion-inherent challenges and enhances motion dataset reliability.

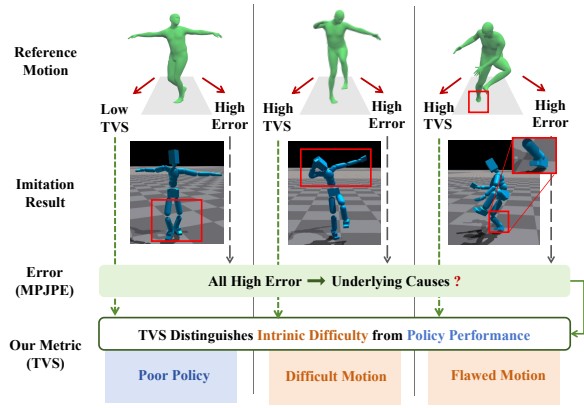

*Figure 1.* Existing metrics assign high error to all three scenarios above equally. In contrast, our Torque Variation Score (TVS) quantifies intrinsic difficulty based on rigid-body dynamics. This allows us to attribute error correctly: high error on low-TVS motions exposes policy limitations (left), whereas high error on high-TVS motions reflects intrinsic difficulty (center and right).

## 1. Introduction

Physics-based motion imitation learning generates physically plausible human-like motions and serves as a cornerstone for humanoid robotics (Luo et al., 2021; 2023a;b). In simulation, reinforcement learning (RL) policies use human motion capture data to drive humanoid avatars, taking a critical step toward real-world deployment. While methods like UHC (Luo et al., 2021) and PHC+ (Luo et al., 2023a) achieve impressive results on large motion datasets, performance remains strikingly inconsistent: under identical training protocols, some motions are replicated with high fidelity, while others exhibit large errors or catastrophic failures. This inconsistency raises a fundamental diagnostic question: **when imitation error occurs, does it stem from policy limitations or the intrinsic learning difficulty of the target motion?** Current evaluation metrics (e.g., MPJPE) only measure final tracking error, conflating these distinct sources. Without quantifying a motion's inherent learning difficulty independently of policy performance, researchers cannot reliably diagnose error origins, hindering fair comparison, targeted improvement, and meaningful progress assessment in imitation learning.

To address this, we propose the Torque Variation Score

---

[1]School of Informatics, Xiamen University, Xiamen, China [2]School of Computer Science and Informatics, Cardiff University, Cardiff, United Kingdom. Correspondence to: Anjun Chen <anjunchen@xmu.edu.cn>.

*Proceedings of the 43$^{rd}$ International Conference on Machine Learning*, Seoul, South Korea. PMLR 306, 2026. Copyright 2026 by the author(s).

(TVS), a physics-grounded metric that quantifies the intrinsic learning difficulty of a motion based solely on its dynamical properties. Derived from rigid-body dynamics, TVS measures the torque variation required to correct small pose perturbations. Intuitively, motions demanding large torque adjustments for minor corrections reside in regions of flat reward landscapes, where standard RL exploration yields negligible state changes and uninformative gradients, making them inherently harder to learn. TVS aggregates the volume, variance, and temporal structure of these torque adjustments to capture this sensitivity comprehensively. We establish a theoretical link between TVS and RL optimization: high-TV motions induce a "vanishing gradient" effect where action noise produces minimal state deviation, starving the policy of learning signals. This explains persistent failures on dynamically sensitive motions (e.g., single-leg stances) versus robust learning on tolerant motions (e.g., hand waving). Crucially, TVS operates *independently of any policy* as it characterizes the motion's learning landscape itself.

We validate TVS through extensive experiments with state-of-the-art policies (UHC (Luo et al., 2021), PHC+ (Luo et al., 2023a)) on standard datasets (AMASS (Mahmood et al., 2019)). TVS strongly correlates with imitation error across diverse motions and enables principled error attribution: high error on low-TV motions signals policy deficiency; high error on high-TV motions reflects fundamental learning constraints imposed by the motion.

As practical demonstrations of TVS's diagnostic utility, we present three applications: *Maximum Imitable Difficulty (MID)* identifies the hardest motions a policy can reliably learn; *Difficulty-Stratified Joint Error (DSJE)* reveals policy strengths/weaknesses across difficulty tiers; and *Motion Anomaly Detection* leverages TVS spikes to flag motion segments with abnormally high learning difficulty (e.g., caused by marker occlusion, self-occlusion, or sensor artifacts), directly supporting mocap data curation and quality assurance. We publicly release TVS scores for major motion datasets to facilitate community adoption and diagnostic analysis. In summary, our contributions are as follow:

- A physics-grounded, policy-agnostic metric that quantifies the intrinsic learning difficulty of motion sequences through torque sensitivity analysis derived from rigid-body dynamics, which we call **Torque Variation Score (TVS)**

- We establish a theoretical connection between TVS and RL optimization challenges (flat reward landscapes, vanishing gradients), empirically validate its strong correlation with imitation error across policies and datasets, and enable principled error attribution between policy limitations and motion-inherent difficulty.

- We demonstrate TVS's practical utility through three diagnostic applications: (1) *Maximum Imitable Difficulty (MID)* for policy capability benchmarking, (2) *Difficulty-Stratified Joint Error (DSJE)* for granular performance profiling, and (3) *Motion Anomaly Detection* for identifying problematic motion segments in motion capture datasets to enhance MoCap data quality and curation workflows.

## 2. Related Works

### 2.1. Physics-based Human Motion Imitation

Physics-based human motion imitation aims to create simulated characters governed by physical laws (Bergamin et al., 2019; Chentanez et al., 2018; Fussell et al., 2021; Gong et al., 2022; Hasenclever et al., 2020; Merel et al., 2020; Peng et al., 2017; 2018a; 2019; 2021; 2022; Wang et al., 2020; Winkler et al., 2022; Yuan & Kitani, 2020; Dou et al., 2023; Yao et al., 2022; 2024), enabling the generation of dynamically plausible human behaviors. Due to the well-known generalization challenges in reinforcement learning based controllers, early efforts focused on small-scale tasks, such as interactive control via user input (Bergamin et al., 2019; Peng et al., 2022; 2021; Wang et al., 2020) or modular skills like locomotion (Peng et al., 2017) and dribbling (Peng et al., 2021). Recent advances have shifted toward large-scale datasets and multi-sequence imitation. ScaDiver (Won et al., 2020) pioneered large-scale imitation of the CMU MoCap dataset (Carnegie Mellon University) using a mixture-of-experts framework, while MoCapAct (Wagener et al., 2022) distilled per-clip experts into a unified policy with minimal performance loss. Most recently, universal motion imitation using a single policy has emerged as a promising direction toward general humanoid control. UHC (Luo et al., 2021) successfully imitates 97% of the AMASS dataset by introducing residual root forces (Yuan & Kitani, 2019) for balance recovery. Its successor, PHC (Luo et al., 2023a), eliminates all non-physical forces, achieving superior physical fidelity while preserving high tracking accuracy. Related works also explore reference tracking (Chao et al., 2021; Chentanez et al., 2018; Park et al., 2019; Peng et al., 2018a;b; Won et al., 2020; Yuan & Kitani, 2020) and object interaction (Chao et al., 2021; Merel et al., 2020) in simulation. Among these methods, UHC and PHC stand out for their ability to handle diverse daily motions and interactions without task-specific tuning, demonstrating robust generalization. Despite these advances, existing evaluations and reward designs rely almost exclusively on similarity-to-reference metrics (e.g., joint position error) across heterogeneous motion datasets, overlooking intrinsic motion properties such as dynamic stability and control fragility.

## 2.2. Human Motion Benchmark

Human motion data serves as a foundational resource for humanoid control, character animation, motion capture, and related applications. In recent years, the widespread adoption of unified parametric models (Loper et al., 2023; Pavlakos et al., 2019) has enabled large-scale datasets to be standardized and broadly applied across motion research. To address limitations in data scale and diversity, recent efforts have turned to internet videos as a rich source for constructing new datasets (Lin et al., 2023; Zhang et al., 2025b; Chung et al., 2021; Cai et al., 2022; Tsuchida et al., 2019). Humanoid-X (Mao et al., 2025) stands out by integrating large-scale video-reconstructed motions with high-fidelity motion capture subsets, creating a hybrid dataset of unprecedented scope. PHUMA (Lee et al., 2025) aim to enhance physical plausibility. Meanwhile, datasets such as (Carnegie Mellon University; Zhang et al., 2022; Al-Hafez et al., 2023) are widely adopted due to their high kinematic accuracy and rich action variety, though constrained by complex camera calibration and marker-based setups. For tasks like general motion imitation and physics-based motion reconstruction, large volumes of uniformly represented data are essential. H36M (Ionescu et al., 2014) provides extensive video-captured human motion, while the widely adapted AMASS dataset (Mahmood et al., 2019) unifies diverse sources under the SMPL parameterization. The community has also produced specialized datasets: (Tsuchida et al., 2019; Li et al., 2023; Aristidou et al., 2019) focus on dance by curating motion sequences synchronized with musical beats—a core aspect of expressive human movement, while BMLHandball (Helm et al., 2017) builds a database of penalty throw actions. These efforts reflect a growing demand for task-specific motion diversity. However, most existing approaches collect data based on semantic categories (e.g., "dance" vs. "locomotion"). While practitioners observe that dance motions are harder to imitate than daily actions, no analysis quantifies the intrinsic difficulty of individual motions, nor explains *why* certain motions are harder. As a result, difficulty-aware applications and in-depth evaluations remain largely unexplored.

## 3. Motivation and Preliminaries

Physics-based human motion imitation leverages physically simulated environments to drive a torque-controlled avatar in mimicking a reference motion. A central challenge in this process is the avatar's propensity to deviate from the reference, often resulting in stumbles or catastrophic falls. To address this, the research community has explored numerous strategies, including RL reward design (Peng et al., 2017; 2021), policy architecture innovation (Won et al., 2020; Luo et al., 2023a), and motion preprocessing (Zhang et al., 2025a), all unified by the goal of maximizing simi-larity to the reference, irrespective of the reference motion itself. Yet, *tracking fidelity is not solely a function of policy capability; it is co-determined by the intrinsic difficulty of the motion*. Empirically, a well-trained policy exhibits striking performance variance across training motions (e.g., joint position error ranging from ∼16 mm on static T-pose to over 150 mm on dynamic motions such as long jump) despite identical training conditions. Therefore, explicitly quantifying motion difficulty is pivotal: it forms the foundation for fairer policy evaluation, difficulty-aware curriculum learning, and physics-plausible restoration of flawed motions, opening new research frontiers in imitation learning. In this work, we aim to fill this gap by introducing a new metric, orthogonal to motion accuracy, that defines and quantifies motion difficulty.

**Problem Setup and Notations.** Our approach is grounded in rigid-body dynamics. We adopt the standard human motion imitation setup with a torque-controlled, floating-base humanoid (Zheng & Yamane, 2013) and follow the mass distribution protocol from (Shimada et al., 2020). We define key relevant 3D human motion properties, including:

- *Joint rotations* (i.e., pose) $\theta \in \mathbb{R}^{3J}$ represented by Euler angles, with their time derivatives $\dot{\theta}$ and $\ddot{\theta}$ representing the angular velocity and angular acceleration;

- *Global joint positions* $r \in \mathbb{R}^{3J}$ with their time derivatives $\dot{r}$ and $\ddot{r}$ representing the linear velocity and acceleration in the global coordinate system, respectively.

We use $J = 24$ to denote the number of joints in the SMPL model. Furthermore, we have:

- *Character configuration* $q = [r_{\text{root}}, \theta] \in \mathbb{R}^N$ defined by its local pose and global translation, where $r_{\text{root}} \in \mathbb{R}^3$ is global translation of the root joint and $N = 3 + 3J$ is the number of degrees of freedom (DoF). The time derivatives $\dot{q}$ and $\ddot{q}$ represent the generalized velocity and acceleration, respectively.

**Equation of Rigid-Body Motion.** In rigid-body dynamics, we define the torque vector $\tau \in \mathbb{R}^N$ for the controlled character, where each dimension corresponds to the force applied to the respective degree of freedom. Then, the character configuration $q$, the torque $\tau$, the generalized velocity $\dot{q}$ and acceleration $\ddot{q}$, satisfy the following equation (Featherstone, 2008):

$$M(q)\ddot{q} + h(q, \dot{q}) = \tau + f_{ext} \quad (1)$$

where $M \in \mathbb{R}^{N \times N}$ is the inertia matrix; $h \in \mathbb{R}^N$ is the nonlinear term accounting for gravity, Coriolis, and centrifugal forces; and $f_{ext}$ denotes the external forces from the surrounding environment of the character.

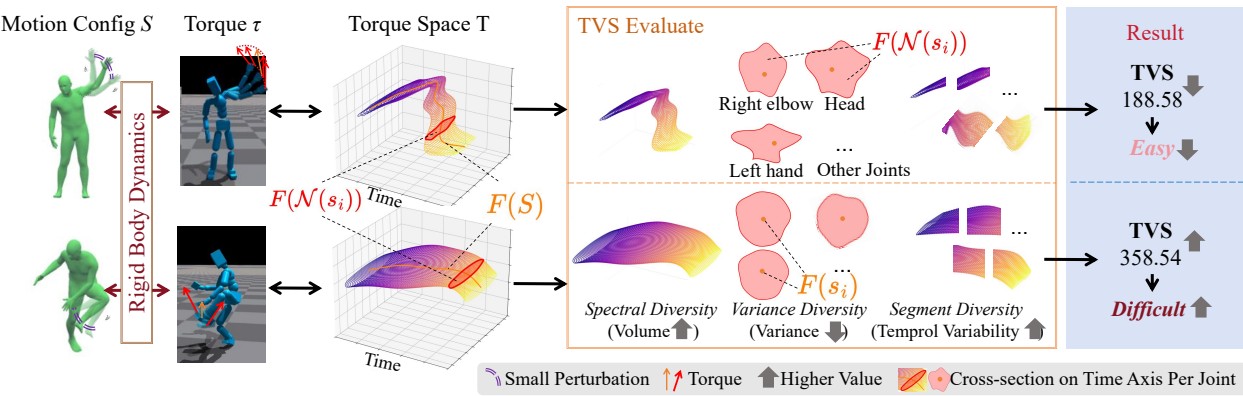

*Figure 2.* Illustration of TVS: For an *easy* motion (top-left), small pose perturbations induce small torque variance and hence low sensitivity to perturbation, the 1) smaller torque space volume, 2) larger variation across joints (larger volume on the waving hand joints than others) and 3) larger temporal variability makes TVS rates the motion as easier. In contract, for a a *difficult* motion (bottom-left), the same level of perturbation yields large torque variance across all joints, leading to a high TVS.

# 4. The Torque Variation Score (TVS) for Quantifying Motion Learning Difficulty

The Torque Variation Score (TVS) provides a physics-grounded measure of the *intrinsic learning difficulty* inherent to a motion sequence—defined as the challenge a motion poses to reinforcement learning optimization, independent of any specific policy. In this section, we formalize TVS from rigid-body dynamics (Sec. 4.1), detail its computation (Sec. 4.2), and describe its application to standard motion datasets (Sec. 4.3).

## 4.1. Quantify Motion Learning Difficulty via Torque Variation

From rigid-body dynamics (Eq. 1), it can be observed that for any state $q$, a perturbation in $q$ induces a corresponding change in the torque space. Therefore, we assume that Motion Learning Difficulty can be quantify via motion's Torque Variation: *The success of imitation learning hinges on how strongly the required torques vary under small motion perturbations.*

Specifically, when tiny pose deviations trigger large torque changes, reinforcement learning (and gradient-based optimization) encounters flat or plateau-like reward (and loss) landscapes: substantial torque updates produce nearly identical poses and thus yield nearly identical rewards. In such regions, the reward becomes weakly informative with respect to torque adjustments, providing very little useful improvement signals.

Intuitively, as illustrated in Fig. 2, for an easy motion such as a hand wave (Fig. 2 top left), a small positional change induces only a mild torque adjustment, and the policy receives clear improvement signals. For a difficult motion such as a single-leg stance (Fig. 2 bottom left), the reward

landscape collapses into a Dirac-delta–like spike, where reward is concentrated within an extremely narrow region and flattens almost immediately elsewhere. In this regime, the policy explores torque updates in many directions on the plateau, yet the resulting rewards are nearly identical.

We therefore define the Torque Variation of motion sequence $S$ as the magnitude of torque variation induced within a bounded pose-error neighborhood:

**Definition 4.1** (Torque Variation of Motion)**.** Let $s = (q, \dot{q}, \ddot{q})$ denote per-frame state. For motion sequence $S = (s_1, \ldots, s_t)$, let $\mathcal{N}(S)$ be a bounded neighborhood of pose-perturbed sequences. The torque variations induced are $\mathcal{T} = \{T \mid T = F(\hat{S}), \hat{S} \in \mathcal{N}(S)\}$. The **intrinsic learning difficulty** of $S$ is characterized by 1) **Volume**: Total magnitude of torque variation; 2) **Variance**: Spread of torque adjustments across joints; 3) **Temporal variability**: Evolution of torque sensitivity over time.

## 4.2. Computing the Torque Variation Score (TVS)

TVS is quantification of Torque Variation that aggregates three complementary proxies reflecting Definition 4.1 (see Fig. 2 for intuition):

### 4.2.1. SPECTRAL DIVERSITY

For a given motion sequence $S$, spectral diversity is used to model the volume $\text{vol}_Y(\mathcal{T})$, *i.e.*, the Lebesgue measure in torque space. By applying the coarea formula and normalizing by $\text{vol}_X(\mathcal{N}(S))$, we have :

$$\text{vol}_Y(\mathcal{T}) \propto \left[\text{vol}_X(\mathcal{N}(S))\right]^{1/3} \cdot \prod_{i=1}^{t} \sqrt{\det(G_i)}, \quad (2)$$

where $G_i = dF_i(s_i) \, dF_i(s_i)^T$. Thus, the volume $\text{vol}_Y(\mathcal{T})$ is determined by $\prod_{i=1}^{t} \sqrt{\det(G_i)}$.

Summing over all frames, the log-volume of the total torque variation is:

$$\log \mathrm{vol}_Y(T) \propto \sum_{i=1}^{r} \log \sigma_i. \tag{3}$$

where $r = \min(t, J \cdot D)$. This shows that the total torque variation grows with the sum of log-singular values across all joints and frames.

We intentionally omit ground contact forces, as they are mathematically negligible for the daily human motions we study. Specifically, from the rigid-body dynamics equation, when computing the torque sensitivity via the Jacobian matrix $J = dF$, the formulation requires computing the derivatives of the internal dynamics terms $(dM, dh)$; and the external contact force $(df_{ext})$. According to (Winter, 2009), external ground reaction forces $f_{ext}$ during daily motions are predominantly governed by Center of Mass (CoM) kinematics and gravity. Because small local pose perturbations do not significantly alter the global CoM trajectory, the variation in external contact forces $df_{ext}$ is infinitesimally small and mathematically negligible compared to $dM$ and $dh$. To provide an intuitive quantitative comparison, we validate this using the real-world dataset GroundLink (Han et al., 2023). During a walking motion, the peak variation of external forces ($|\frac{\partial f_{ext}}{\partial q}|$) is merely 2.4. Even for highly dynamic motions with larger impact frequency, such as hopping, this peak only reaches 7.9. Conversely, on a regular walking motion, $|\frac{\partial (M\ddot{q})}{\partial q}|$ averages 1046, and $|\frac{\partial h}{\partial q}|$ peaks at 1288. This extreme ratio ( 0.23%) demonstrates that $df_{ext}$ is mathematically safe to ignore. While one might hypothesize extreme theoretical scenarios where massive, discontinuous contact forces could influence the computation, our observations suggest such cases are likely non-existent in natural human biomechanics.

Thus, we define **Spectral Diversity** as:

$$d_1 = \sum_{i=1}^{r} (\log \sigma_i), \tag{4}$$

Spectral Diversity thus captures the effective dimensionality of the feasible torque space, higher values indicating higher torque variation and higher difficulty. **Due to space constraints, we provide the detailed derivation for the log-volume of torque variation in the Appendix**.

### 4.2.2. VARIANCE DIVERSITY

Eq. 18 shows that the volume of the torque space is directly related to $G_i$ , hence to the Jacobian matrix $J = dF_i$. Therefore, we define **Variance Diversity** as the variation of the Jacobian across different joints:

$$d_2 = \sum_{j=1}^{J} \log(\mathrm{Var}_j), \tag{5}$$

where:

$$\mathrm{Var}_j = \mathbb{E}_{t,k}[\mathbf{J}_{t,j,k}^2] - (\mathbb{E}_{t,k}[\mathbf{J}_{t,j,k}])^2, \tag{6}$$

and $t$ is the time index, $k$ is the index over rotation directions. The variance $\mathrm{Var}_j$ quantifies the spread in torque variation for joint $j$, and $d_2$ aggregates this across all joints via a geometric mean. Because variance in the Jacobian entries corresponds to the magnitude of singular values, Variance Diversity captures the extent of joint-wise torque variation, where smaller variance signifies higher motion difficulty.

### 4.2.3. SEGMENT DIVERSITY

To capture temporal evolution of torque variations, we partition the motion sequence into $K$ non-overlapping subsequences and compute **Segment Diversity** as the sum of spectral diversity per segment:

$$d_3 = \frac{1}{K} \sum_{k=1}^{K} d_1(J_k) \propto \frac{1}{K} \sum_{k=1}^{K} \log V_s = \log \left( \prod_{s=1}^{S} V_s \right)^{1/S}, \tag{7}$$

where $J_k$ is the Jacobian sequence in segment $k$. From inequality of arithmetic and geometric mean, we have:

$$d_3 \propto \log \left( \prod_{s=1}^{S} V_s \right)^{1/S} \leq \frac{1}{S} \sum_{s=1}^{S} V_s. \tag{8}$$

The upper bound of the inequality is attained if and only if $d_1(J_1) = d_1(J_2) = \cdots = d_1(J_K)$, at which point the segment diversity reaches its maximum value. This means that for a motion sequence $S$ with fixed total torque volume, $d_3$ is maximized when all segments have equal volume, hence more difficult to imitate, consistent with the *Temporal Variability* factor. Conversely, $d_3$ decreases as the volume disparity between segments grows. Intuitively, a motion containing both relatively easy and difficult segments can facilitate learning because these segments are correlated: progress on the easier portions provides intermediate structure that guides the policy toward successfully handling the more challenging parts. For instance, in a kicking motion, the initial weight-shifting and leg-lifting phases are relatively easy, while the final fast extension and balance recovery are more difficult. As the policy masters the easier early phases, it gains stable trajectories and meaningful reward signals that guide it toward successfully learning the harder components. Empirically, we set $K = 4$ in our experiments to effectively capture temporal dynamics.

### 4.2.4. FINAL FORMULATION

The final Torque Variation Score (TVS) is computed as the weighted sum of the three diversity terms, the weight

*Table 1.* Correlation between TVS and imitation error (MPJPE-G) across policies. Higher TVS consistently predicts higher error, validating TVS as a measure of intrinsic learning difficulty. Representative samples illustrate the TVS-error relationship.

| | UHC | | | | | | PHC+ | | | | | |
|---|---|---|---|---|---|---|---|---|---|---|---|---|
| **Correlation** | **Pearson** 0.4823 | | **Spearman** 0.6586 | | **Kendall** 0.4747 | | **Pearson** 0.6478 | | **Spearman** 0.8203 | | **Kendall** 0.6293 | |
| | **Motion** | **TVS** | **MPJPE-G** | **Motion** | **TVS** | **MPJPE-G** | **Motion** | **TVS** | **MPJPE-G** | **Motion** | **TVS** | **MPJPE-G** |
| **Samples** | hand_waving | 188.58 | 16.67 | tpose | 215.71 | 16.34 | walk_slow | 209.82 | 19.75 | touch_head | 229.73 | 36.39 |
| | tpose2 | 267.20 | 23.29 | throw | 259.71 | 25.28 | wave | 225.34 | 41.23 | right_punch | 227.89 | 43.77 |
| | tennis | 273.48 | 26.20 | twist | 270.28 | 29.77 | spin | 257.09 | 55.98 | wipe | 266.71 | 44.28 |
| | small_jump | 309.36 | 38.21 | dodge | 325.62 | 41.42 | throw | 259.71 | 56.15 | small_step | 280.27 | 50.23 |
| | walk | 337.27 | 48.20 | kick | 328.67 | 55.56 | reach_high | 304.39 | 60.17 | punch | 318.90 | 53.17 |
| | dodge2 | 342.72 | 48.45 | jump | 333.87 | 65.31 | stretch | 317.90 | 60.52 | walk | 337.27 | 65.85 |
| | spin2 | 355.95 | 62.57 | run | 358.97 | 61.47 | run | 355.76 | 61.67 | kick | 328.67 | 73.65 |
| | hop | 358.54 | 63.40 | spin2 | 355.95 | 62.56 | run_fast | 349.02 | 103.23 | hop | 358.54 | 73.85 |
| | fast_jump | 316.30 | 130.68 | run_fast | 349.02 | 150.21 | spin2 | 355.95 | 70.10 | fast_spin | 372.48 | 75.00 |

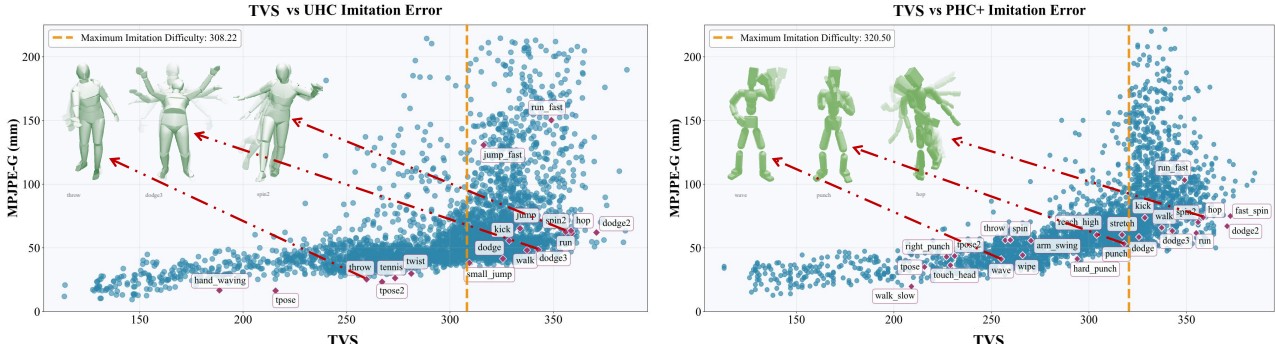

*Figure 3.* We plot scatters of TVS versus imitation error on different polices for over 3000 motion clips (left: UHC; right: PHC+), with samples drawn from the policies' training set. As TVS increases from left to right (indicating higher difficulty), error rises from bottom to top, demonstrating that TVS effectively captures motion difficulty.

assignments are empirically derived:

$$\text{TVS} = \sum_{i=1}^{3} w_i d_i, \tag{9}$$

We choose $\omega_1, \omega_2, \omega_3 = [0.4, 0.3, 0.3]$ in our implementations.

**Detailed hyperparameter analyses, including $\omega_1, \omega_2, \omega_3$ and the segment count $K$, are provided in the Appendix.**

### 4.3. TVS-annotated Motion Dataset

To facilitate community adoption and diagnostic analysis, we compute TVS for all short clips (100 frames) in the AMASS dataset (Mahmood et al., 2019), following standard segmentation practices in imitation learning. This yields **TVS-annotated AMASS**: over 30,000 clips each labeled with a scalar TVS value. Crucially, this is *not* a new benchmark dataset but a *difficulty annotation* of an existing resource. The modular TVS computation pipeline is publicly released to enable annotation of any motion dataset (e.g., AIST++). This supports—rather than prescribes—difficulty-aware analysis.

## 5. Empirical Validation of TVS

In this section, we validate TVS not as a benchmark, but as a diagnostic measure of learning difficulty through correlation with policy errors and ablation studies. Our experiments reveal a strong positive correlation between TVS and policy errors, confirming that higher TVS scores accurately predict greater difficulty in motion imitation.

### 5.1. Correlation with Imitation Error

Using TVS-annotated AMASS, we evaluate state-of-the-art policies (UHC, PHC+) on over 3,000 motion clips. Fig. 3 shows strong positive correlation between TVS and MPJPE-G (e.g. global mean per-joint position error): higher TVS consistently predicts higher imitation error. Table. 1 reports Pearson, Spearman, and Kendall correlations—all significantly positive—confirming TVS captures intrinsic learning difficulty. Correlation strength is typically interpreted using standardized effect size conventions rather than by direct comparison of raw values. For example, according to the established statistical guidelines by Jacob Cohen in Statistical Power Analysis for the Behavioral Sciences, a

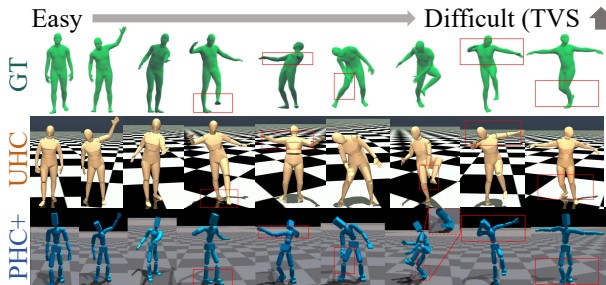

*Figure 4.* Imitation fidelity degrades as TVS increases. Low-TV motions (left) are replicated accurately; high-TV motions (right) exhibit errors where torque sensitivity impedes learning.

Spearman correlation of r = 0.10 is considered "Small", r = 0.30 is "Medium", and r ≥ 0.50 is "Large/Strong". In our experiments, the Spearman correlations for both policies consistently exceed 0.60 (0.658 for UHC and 0.820 for PHC+), demonstrating a robust and strong correlation across the board. In other words, the observed "strong correlation" between TVS and imitation errors is insensitive to the specific policy used. Qualitatively (Fig. 4), low-TV motions (e.g., hand wave) are imitated accurately; high-TV motions (e.g., fast spin) show clear degradation where policies trade precision for stability due to uninformative gradients.

**Due to space constraint, we provide more detailed validation results with over 10000 samples in the Appendix.**

### 5.2. Ablation Study

Table 2 shows ablation results. Removing any component (Spectral/SP, Variance/VA, Segment/SE diversity) reduces correlation with policy error. Full TVS achieves highest scores across all metrics and policies, empirically confirming that all three factors are necessary to capture motion learning difficulty comprehensively.

### 5.3. Empirical Evidence for "Flat Reward Landscapes"

To provide empirical evidence for *why* TVS quantifies motion imitation difficulty, we evaluate the reward sensitivity of a pre-trained policy to verify the theoretical claim of a flat reward landscape, aiming to validate our theory that high Torque Variation (TVS) motions induce "flat reward landscape". Specifically, we apply progressively increasing random pose perturbations (denoted as Perturbation levels 1 < 2 < 3) to motions of varying difficulty and track the corresponding reward. The results are summarized in Table 3. For Low-TVS (easy) motions, the reward drops significantly and monotonically as the perturbation magnitude increases. This steep gradient in the reward landscape provides a clear, informative learning signal, effectively guiding the policy toward the correct reference pose. In contrast, for High-TVS (difficult) motions, the reward landscape collapses into a

*Table 2.* Ablation study: Correlation of partial TVS formulations with imitation error. Full TVS achieves highest correlation, validating the necessity of all three components.

| Score | Pearson | Spearman | Kendall |
|---|---|---|---|
| **UHC** | | | |
| *w/o VA* | 0.4455 | 0.6218 | 0.4443 |
| *w/o SE* | 0.4475 | 0.6231 | 0.4455 |
| *w/o SP* | 0.4126 | 0.5847 | 0.4149 |
| TVS (full) | **0.4823** | **0.6586** | **0.4747** |
| **PHC+** | | | |
| *w/o VA* | 0.6006 | 0.7963 | 0.6005 |
| *w/o SE* | 0.6026 | 0.7966 | 0.6014 |
| *w/o SP* | 0.5760 | 0.7803 | 0.5867 |
| TVS (full) | **0.6478** | **0.8203** | **0.6293** |

*Table 3.* Reward responses of the pre-trained policy (PHC+) under progressively increasing pose perturbations for Low-TVS and High-TVS motions. High-TVS motions exhibit reward plateauing, empirically validating the flat optimization landscape claim.

| Perturbation Level | Low TVS ($< 150$) | | High TVS ($> 350$) | |
|---|---|---|---|---|
| | Reward | Δ Reward | Reward | Δ Reward |
| No Perturbation | 263.99 | - | 190.63 | - |
| Perturbation 1 | 253.70 | -10.29 | 187.63 | -3.00 |
| Perturbation 2 | 247.46 | -6.24 | 187.23 | -0.40 |
| Perturbation 3 | 224.75 | -22.71 | 187.74 | +0.51 |

plateau. Despite increasing perturbation magnitudes, the reward remains nearly static (hovering around ∼187), exhibiting negligible variance. Consequently, as the policy explores drastically different perturbed states, it receives virtually indistinguishable rewards. This lack of differential feedback induces a vanishing gradient effect, starving the policy of actionable learning signals. These empirical measurements validate our theoretical assertion that high dynamic sensitivity (high TVS) induces flat reward landscapes, fundamentally bottlenecking the imitation learning optimization process.

## 6. Diagnostic Applications of TVS

TVS enables principled error attribution. In this section, we present three analyses as illustrative demonstrations of its diagnostic utility,

### 6.1. Maximum Imitable Difficulty (MID) as a Diagnostic Illustration

Revisiting Fig. 3, we observe that while both UHC and PHC+ exhibit a strong overall positive correlation with TVS, a growing number of outliers emerge as difficulty increases. These are particularly evident on the right side of the scatter plots, where imitation error surges dramatically for high-TVS samples. The pattern is also reflected in Ta-

*Table 4.* Difficulty-Stratified Joint Error (DSJE) at fixed TVS thresholds (mm).

| Policy | DSJE-200 | DSJE-300 | DSJE-350 |
|--------|----------|----------|----------|
| UHC    | **27.79** | 45.75    | 62.94    |
| PHC+   | 30.38    | **39.01** | **59.50** |

ble 1 (e.g., run_fast). Such outliers signify a *breakdown in policy capability* on challenging motions, marking the onset of performance collapse.

To quantify this transition, we exhaustively evaluate all possible TVS split points (e.g., 100, 101, ..., up to the maximum) and, for each, compute the mean MPJPE-G of the left (low-difficulty) and right (high-difficulty) partitions. The split that *maximizes inter-group error disparity* defines the critical difficulty level at which large errors begin to dominate. We term this threshold the **Maximum Imitable Difficulty (MID)**, a direct measure of a policy's robustness boundary. We formally define it as:

$$\text{MID} = \arg\max_{c \in \mathcal{C}} \left[ \mu_{\text{high}}(c) - \mu_{\text{low}}(c) \right], \qquad (10)$$

where:

- $\mathcal{C}$ is the set of all candidate TVS thresholds;

- $\mathcal{D}_{\text{low}} = \{S \in \mathcal{S} | \text{TVS}(S)) \leq c\}$ and $\mathcal{D}_{\text{high}} = \{S \in \mathcal{S} | \text{TVS}(S)) > c\}$ are the low- and high-difficulty partitions, $\mathcal{S}$ is set of all motion clips and $\text{TVS}(S)$ is TVS of $S$;

- $\mu_{\text{low}}(c) = \frac{1}{|\mathcal{D}_{\text{low}}|} \sum_{S \in \mathcal{D}_{\text{low}}} \text{MPJPE-G}(S))$ is the mean imitation error for motions with TVS $\leq c$, MPJPE-G$(S)$ is MPJPE-G of $S$;

- $\mu_{\text{high}}(c) = \frac{1}{|\mathcal{D}_{\text{high}}|} \sum_{S \in \mathcal{D}_{\text{high}}} \text{MPJPE-G}(S))$ is the mean imitation error for motions with TVS $> c$.

The MID thus identifies the precise TVS value where the *marginal increase in imitation error* is maximized, marking the boundary beyond which the policy's performance degrades substantially. In Fig. 3, we mark the computed MID with red dashed lines: UHC achieves MID = 308.22, while PHC+ reaches MID = 320.50. This quantitatively confirms PHC+'s superior imitation capacity, with UHC exhibiting earlier onset of outliers, which is a distinction clearly visible in the plots.

**We provide additional validation on the proposed MID metric in the Appendix.**

## 6.2. Difficulty-Stratified Error (DSJE) Analysis

Existing evaluation protocols typically report mean policy error across the entire dataset. While this provides a useful summary of overall performance, we argue that it obscures critical insights: a lower mean error does not guarantee superiority on *every* motion, nor does it reveal where a policy truly excels or falters. To address this limitation, we introduce a difficulty-stratified evaluation using our proposed benchmark. Specifically, we define the Difficulty-Stratified Joint Error (DSJE) at increasing difficulty thresholds:

- DSJE-200: error on easy motions (TVS < 200),

- DSJE-300: error on moderate motions (TVS < 300),

- DSJE-350: error on all motions up to high difficulty (TVS < 350).

The unified formulation for DSJE at threshold $c$ is:

$$\text{DSJE-}c = \frac{1}{|\mathcal{M}_c|} \sum_{S \in \mathcal{M}_c} \text{MPJPE-G}(S), \qquad (11)$$

where $\mathcal{M}_c = \{S \in \mathcal{S} | \text{TVS}(S) < c\}$, $|\mathcal{M}_c|$ denotes the number of motions in the difficulty-stratified subset.

These metrics are reported in Table 4 for both UHC and PHC+. While conventional aggregate evaluation confirms PHC+'s overall superiority over UHC, our DSJE analysis reveals a surprising counter-pattern: PHC+ underperforms UHC on the easiest motions (DSJE-200), despite outperforming on moderate and cumulative sets. This novel finding emerged only through our difficulty-aware lens, and offers deeper diagnostic value. It enables precise characterization of policy behavior, training dynamics, and task-specific applicability: *which model is best suited for which difficulty regime?* Such insights pave the way for targeted improvements, curriculum design, and principled deployment in real-world humanoid control.

## 6.3. Flawed Motion Detection

Motion capture (mocap) data—regardless of acquisition modality—frequently suffer from technical artifacts and environmental interference, yielding physically implausible or noisy motions that compromise downstream applications. Optical systems commonly encounter marker occlusion (due to self-occlusion, clothing, or environmental obstacles) and marker dropout, introducing trajectory gaps and kinematic inconsistencies. Meanwhile, IMU-based systems face distinct challenges including sensor noise, drift, and calibration errors (Zuo et al., 2025; Huang et al., 2018; Shao et al., 2025). These modality-specific failure modes underscore the critical need for robust preprocessing, physics-based

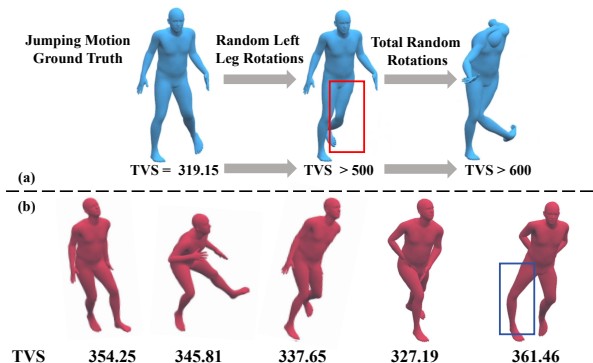

*Figure 5.* Illustration of the flawed motion detection potential. **(a)** As an increasing number of joints in an originally natural motion are replaced with random, physically implausible rotations (i.e., introducing "flawed" poses), the TVS score rises dramatically. **(b)** On real noisy mocap data, visibly flawed motions (e.g., globally unbalanced or locally unnatural poses) exhibit substantially higher TVS values. Samples are picked from out TVS-AMASS (optical-capture) and SAIP dataset (inertial-capture) (Yin et al., 2025).

validation, and error-aware evaluation pipelines in motion analysis workflows.

In this section, we conduct a simple yet illustrative experiment to demonstrate that our proposed TVS assigns markedly higher difficulty scores to unrealistic motions, thereby revealing its potential as an effective detector of such anomalies. We first demonstrate the behavior of TVS on "flawed" motions. For a natural jumping sequence (Fig. 7 (a), left), TVS yields a score of 319. When the four joints of the left leg are perturbed with random rotations (Fig. 7 (a), middle), the score exceeds 500. Further corrupting additional joints with physically implausible random rotations pushes TVS beyond 600 (Fig. 7 (a), right). These results provide clear evidence that unrealistic motions are consistently assigned substantially higher difficulty. Leveraging this property, we manually select several visually anomalous sequences from a representative IMU-based mocap dataset (Yin et al., 2025) (Fig. 7 (b)). Unstable and unbalanced motions (Fig. 7 (b), left four examples) exhibit significantly elevated TVS values, while sequences containing unnatural limb contortions (Fig. 7 (b), right) yield scores exceeding 360. The consistently high difficulty assigned to such defective motions underscores the potential of TVS as a principled and automated tool for flawed motion detection.

**Additional potential applications of TVS are discussed in the Appendix, including TVS-guided curriculum learning and TVS's generalization to other humanoid robots.**

## 7. Conclusion and Limitations

**Conclusion.** We introduced the Torque Variation Score (TVS), a physics-grounded measure that quantifies the intrinsic learning difficulty of motion sequences from rigid-body dynamics. TVS operates independently of policy performance, enabling principled attribution of imitation errors: high error on low-TV motions indicates policy limitations; high error on high-TV motions reflects fundamental learning constraints. Through extensive validation on state-of-the-art policies, we demonstrated that TVS shows a strong correlation with policy errors and its utility for diagnostic analysis. As illustrative applications, MID and DSJE analyses reveal nuanced policy behaviors unattainable with aggregate metrics. TVS provides a rigorous lens to distinguish policy-induced errors from those inherent to motion learning difficulty, advancing transparent evaluation and targeted improvement in humanoid imitation learning.

**Limitations.** Our current implementation of TVS computation relies on the CPU-based dynamics library RBDL (Felis, 2017), which takes roughly 1–2 minutes to calculate a 100-frame clip on an Intel i9-14900KF. GPU-accelerated dynamics solvers could significantly improve efficiency. Additionally, TVS characterizes difficulty under idealized dynamics; real-world factors (sensor noise, actuator limits) may introduce additional challenges not captured here. We will try to extend TVS to incorporate such practical constraints in future work.

## Acknowledgements

Supported by .This work is supported by National Natural Science Foundation of China (62472364, 62072383), the Public Technology Service Platform Project of Xiamen City (No.3502Z20231043), Xiaomi Young Talents Program / Xiaomi Foundation and the Fundamental Research Funds for the Central Universities (20720240058), "Young Eagle Plan" Top Talents of Fujian Province, Fujian Provincial Natural Science Foundation of China (2026J002001). Anjun Chen is the corresponding author. .

## Impact Statement

This paper presents a metric aimed at advancing the field of physics-based motion imitation and reinforcement learning.

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

# A. Spectral Diversity Details

Following the definitions from the main paper, we complete the detailed proofs and definitions for the relevant propositions regarding spectral diversity, including:

- definition of the torque space volume $\text{vol}_Y(\mathcal{T})$

- the relationship between $\text{vol}_Y(\mathcal{T})$ and the Gram determinant $\det(G_i)$ (Equation 2);

- computation of the $\det(G_i)$.

## A.1. Torque Space Volume Definition

We first revisit the problem setup. Let $S = (s_1, s_2, ..., s_t)$ be a motion sequence of $t$ frames, where the state at frame $i$ is $s_i = (q_i, \dot{q}_i, \ddot{q}_i)$, with $q_i$ denoting position and rotation, $\dot{q}_i$ and $\ddot{q}_i$ are the generalized velocity and acceleration, respectively. Given $\dim q = N$, $F : X \to Y$ maps states $S$ in $X = \mathbb{R}^{N \times 3 \times t}$ to torques $T$ in $Y = \mathbb{R}^{J \times t}$, where $J$ is the number of joints.

Here, $F : X \to Y$ is defined by the rigid-body motion equation described in Section 3 of the main paper:

$$\tau = F(S) = M(q)\ddot{q} + h(q, \dot{q}) - f_{ext}, \tag{12}$$

and for each frame $i$,

$$\tau_i = F_i(s_i) = M(q_i)\ddot{q}_i + h(q_i, \dot{q}_i) - f_{ext}. \tag{13}$$

When $S$ is perturbed, we define the neighborhood $\mathcal{N}(S)$ as the Cartesian product of $\epsilon$-balls centered at each reference state $s_i \in S$:

$$\mathcal{N}(S) = \prod_{i=1}^{t} B_\epsilon(s_i). \tag{14}$$

The image set $\mathcal{T} = F(\mathcal{N}(S))$ is then

$$\mathcal{T} = \prod_{i=1}^{t} F_i(B_\epsilon(s_i)). \tag{15}$$

Volume computation involves the volumes of the state space $X$ and torque space $Y$. Since $q$ includes rotations, $S$ is a manifold, so the volume of a local ball scales as

$$\text{vol}_S(B_\epsilon(s_i)) \propto \epsilon^{3N}, \tag{16}$$

with the constant determined by the induced metric at $s_i$. Consequently,

$$\text{vol}_X(\mathcal{N}(S)) \propto \epsilon^{3Nt}. \tag{17}$$

The volume $\text{vol}_Y(\mathcal{T})$ is the Lebesgue measure in $Y$.

## A.2. Volume Transformation and the Gram Determinant

In this section, we elaborate on Equation (2) introduced in Section 4.2.1 of the main paper and prove the relationship between the torque space volume and the Gram determinant $G_i$.

**Proposition A.1.** *For a given motion sequence $S$, the volume of the torque variation $\mathcal{T}$ induced by $\mathcal{N}(S)$ satisfies:*

$$vol_Y(\mathcal{T}) \propto \left[vol_X(\mathcal{N}(S))\right]^{1/3} \cdot \prod_{i=1}^{t} \sqrt{\det(G_i)}, \tag{18}$$

*where $G_i = dF_i(s_i)\, dF_i(s_i)^T$. Thus, the volume $vol_Y(\mathcal{T})$ is determined by $\prod_{i=1}^{t} \sqrt{\det(G_i)}$.*

*Proof.* For the per-frame mapping $F_i : S \to \mathbb{R}^N$ with $\dim S = 3N > N$, if $F_i$ is a submersion at $s_i$ (i.e., its differential $dF_i(s_i)$ is surjective), for sufficiently small $\epsilon$, the volume of the image set in $\mathbb{R}^N$ is approximated by the coarea formula as

$$\text{vol}_{\mathbb{R}^N}\big(F_i(B_\epsilon(s_i))\big) \approx \text{vol}_S(B_\epsilon(s_i)) \cdot \sqrt{\det\big(dF_i(s_i)\, dF_i(s_i)^T\big)}, \tag{19}$$

where $dF_i(s_i) \in \mathbb{R}^{N \times 3N}$ is the Jacobian of $F_i$ at $s_i$. The Gram matrix $dF_i(s_i)\, dF_i(s_i)^T \in \mathbb{R}^{N \times N}$ encodes local volume distortion, and its determinant's square root governs the scaling factor. Since $\mathcal{T}$ is a product space, the total volume is the product of per-frame contributions:

$$\begin{aligned}
\text{vol}_Y(\mathcal{T}) &\approx \prod_{i=1}^{t} \text{vol}_{\mathbb{R}^N}\big(F_i(B_\epsilon(s_i))\big) \\
&\approx \prod_{i=1}^{t} \Big[ \text{vol}_S(B_\epsilon(s_i)) \cdot \sqrt{\det(G_i)} \Big].
\end{aligned} \tag{20}$$

Substituting the manifold volume scaling $\text{vol}_S(B_\epsilon(s_i)) \propto \epsilon^{3N}$, we obtain

$$\text{vol}_Y(\mathcal{T}) \propto \epsilon^{Nt} \cdot \prod_{i=1}^{t} \sqrt{\det(G_i)}. \tag{21}$$

Given that the state-space neighborhood volume scales as $\text{vol}_X(\mathcal{N}(S)) \propto \epsilon^{3Nt}$, normalizing yields the $\epsilon$-independent relation same as Eq. 18:

$$\text{vol}_Y(\mathcal{T}) \propto \big[\text{vol}_X(\mathcal{N}(S))\big]^{1/3} \cdot \prod_{i=1}^{t} \sqrt{\det(G_i)}. \tag{22}$$

Since $\big[\text{vol}_X(\mathcal{N}(S))\big]^{1/3}$ remains constant for sequences of the same length, $\text{vol}_Y(\mathcal{T})$ is determined by $\prod_{i=1}^{t} \sqrt{\det(G_i)}$. $\quad\square$

### A.3. Computation of the Gram Determinant

**Proposition A.2.** *Derived from Eq. 22, the log-volume of the torque variation is:*

$$\log \text{vol}_Y(T) \propto \sum_{i=1}^{r} \log \sigma_i. \tag{23}$$

*where $r = \min(t, J \cdot D)$, and $D = \dim S = 3N$.*

*Proof.* For each frame, the Jacobian matrix $dF_i \in \mathbb{R}^{N \times D}$ yields a Gram matrix that is an $N \times N$ symmetric positive definite matrix (since the inertial matrix $M(q)$ ensures that $dF_i$ has full row rank). Let the Singular Value Decomposition (SVD) of $dF_i$ be $dF_i = U_i \Sigma_i V_i^T$, where $\Sigma_i = \text{diag}(\sigma_{i,1}, \sigma_{i,2}, \dots, \sigma_{i,N})$, and $\sigma_{i,j} \geq 0$ are the singular values. Accordingly, we have:

$$G_i = U_i \Sigma_i \Sigma_i^T U_i^T = U_i \cdot \text{diag}(\sigma_{i,1}^2, \sigma_{i,2}^2, \dots, \sigma_{i,N}^2) \cdot U_i^T, \tag{24}$$

Therefore, we have:

$$\det(G_i) = \prod_{j=1}^{N} \sigma_{i,j}^2, \quad \sqrt{\det(G_i)} = \prod_{j=1}^{N} \sigma_{i,j}. \tag{25}$$

Taking the logarithm:

$$\log \sqrt{\det(G_i)} = \sum_{j=1}^{N} \log \sigma_{i,j}. \tag{26}$$

Summing over all frames, the log-volume of the total torque variation is:

$$\log \text{vol}_Y(T) \propto \sum_{i=1}^{r} \log \sigma_i. \tag{27}$$

where $r = \min(t, J \cdot D)$.

$\quad\square$

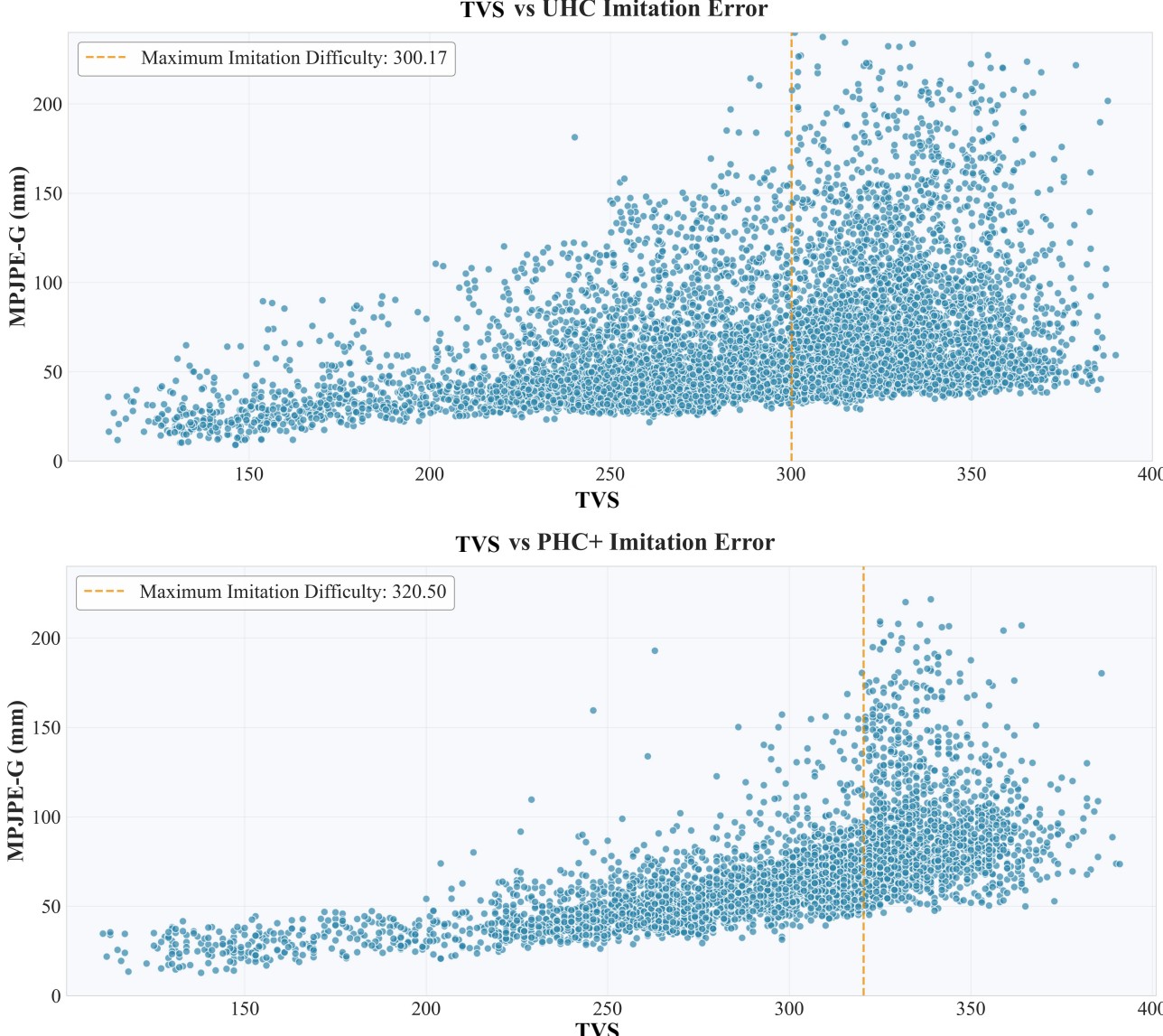

*Figure 6.* More comprehensive scatter plots of TVS vs imitation error (top: UHC; bottom: PHC+), with over 10,000 motion samples drawn from the MD-AMASS dataset, providing stronger statistical evidence of the robust correlation between our proposed TVS and humanoid control error across different policy architectures.

## B. Extensive Validation

### B.1. Imitation Error Correlation

To complement the validation in the main paper, we conduct an extensive evaluation on over 10,000 samples from the MD-AMASS dataset, enabling a more thorough examination of the effectiveness of TVS. Consistent with the validation results in the main paper, motions with higher TVS exhibit weaker policy performance, as shown in Fig. 6.

### B.2. Hyperparameter Analysis

We conduct a sensitivity analysis on the key hyperparameters of TVS: the integration weights $\omega_1, \omega_2, \omega_3$ and the segment count $K$. We evaluate the robustness of our metric by measuring its correlation (Pearson, Spearman, and Kendall) with the UHC imitation error (MPJPE-G) across various hyperparameter configurations.

*Table 5.* Sensitivity analysis of the integration weights $\omega_1, \omega_2, \omega_3$. The correlation between TVS and imitation error remains highly stable even under extreme weight skewness.

| $\omega_1$ | $\omega_2$ | $\omega_3$ | Pearson | Spearman | Kendall |
|---|---|---|---|---|---|
| **0.4** | **0.3** | **0.3** | 0.4823 | **0.6586** | **0.4747** |
| 0.3 | 0.4 | 0.3 | 0.4832 | 0.6502 | 0.4740 |
| 0.3 | 0.4 | 0.4 | **0.4838** | 0.6584 | 0.4745 |
| 0.5 | 0.25 | 0.25 | 0.4760 | 0.6550 | 0.4716 |
| 0.25 | 0.5 | 0.25 | 0.4209 | 0.6233 | 0.4369 |
| 0.25 | 0.25 | 0.5 | 0.4781 | 0.6562 | 0.4725 |
| 0.7 | 0.15 | 0.15 | 0.4657 | 0.6444 | 0.4628 |
| 0.15 | 0.7 | 0.15 | 0.4473 | 0.6172 | 0.4409 |
| 0.15 | 0.15 | 0.7 | 0.4616 | 0.6478 | 0.4653 |

*Table 6.* Sensitivity analysis of the segment count $K$ in Segment Diversity. The overall performance is robust across a wide range of temporal partitioning choices.

| $K$ | Pearson | Spearman | Kendall |
|---|---|---|---|
| 2 | 0.4618 | 0.6524 | 0.4552 |
| 3 | 0.4691 | 0.6534 | 0.4572 |
| **4** | 0.4823 | **0.6586** | **0.4747** |
| 5 | 0.4763 | 0.6534 | 0.4542 |
| 6 | 0.4793 | **0.6586** | 0.4501 |
| 10 | 0.4813 | 0.6554 | 0.4491 |
| 15 | **0.4856** | 0.6516 | 0.4470 |

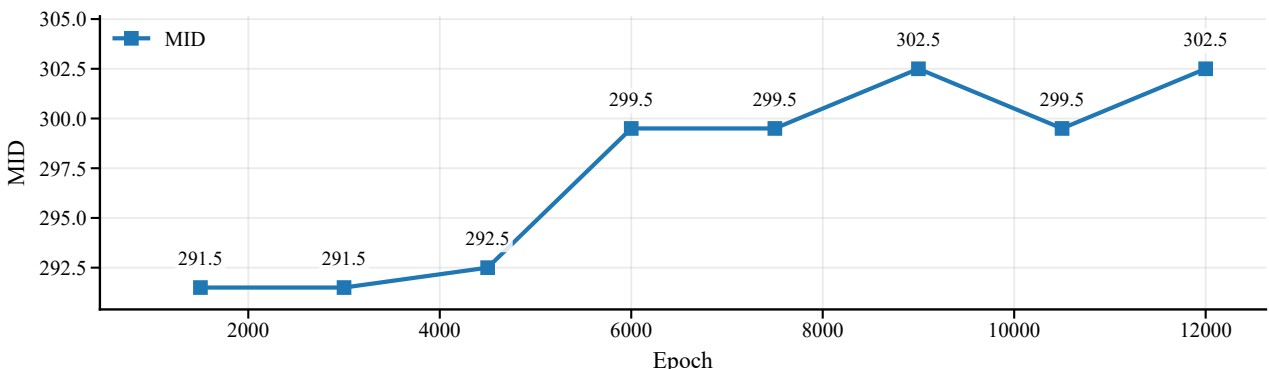

*Figure 7.* We plot displays the MID values calculated for the PHC policy across training steps.

- **Integration Weights** $(\omega_1, \omega_2, \omega_3)$**.** To evaluate the sensitivity of the final score to weight allocation, we assign a dominant weight to one specific component while keeping the other two smaller and equal (e.g., 0.5, 0.25, 0.25). This tests whether over-relying on any single factor degrades the metric's overall correlation. As shown in Table 5, even under highly skewed weight distributions (e.g., assigning up to 0.7 to a single component), the correlation coefficients remain remarkably stable, with the Spearman correlation consistently exceeding 0.61. This empirical evidence substantiates that TVS is fundamentally robust to weight selection. In our implementation, we empirically set $\omega_1 = 0.4$, $\omega_2 = 0.3$, and $\omega_3 = 0.3$.

- **Segment Count** $(K)$**.** We similarly evaluate the sensitivity of the segment count $K$ used in the Segment Diversity computation. Table 6 demonstrates that varying $K$ across a wide range (from 2 to 15) yields minimal fluctuations in correlation performance. This indicates that our method is largely insensitive to the temporal partition granularity. Based on these results, we set $K = 4$.

*Table 7.* Quantitative comparison results between TVS-guided curriculum learning model (Ours) and baseline model (w/o CL).

| Method | MPJPE-G (mm) ↓ | MPJPE-L (mm) ↓ | Acc-Dist (mm/frame²) ↓ | Vel-Dist (mm/frame) ↓ |
|---|---|---|---|---|
| **Overall Performance** | | | | |
| w/o CL | 48.08 | 32.22 | 3.36 | 5.32 |
| Ours | **45.22** | **30.98** | **3.19** | **5.21** |
| **TVS<200** | | | | |
| w/o CL | 36.30 | 25.67 | 1.13 | 1.94 |
| Ours | **31.08** | **23.61** | **1.09** | **1.84** |
| **200<TVS<300** | | | | |
| w/o CL | 49.94 | **35.04** | 3.04 | 4.96 |
| Ours | **49.52** | 35.43 | **2.93** | **4.83** |
| **TVS>300** | | | | |
| w/o CL | 60.79 | 37.44 | 6.32 | 9.65 |
| Ours | **59.18** | **35.90** | **5.87** | **9.52** |

## B.3. MID Validation

To further validate the rationality of MID as a diagnostic tool, we tracked the evolution of the MID score during policy training, hypothesizing that a valid robustness metric should quantitatively reflect the policy's growing capability. We train a PHC (Luo et al., 2023a) policy and evaluate checkpoints at regular intervals of 1,500 steps (ranging from 1,500 to 12,000) on the AMASS dataset. As illustrated in Fig. 7, the MID score exhibits a clear upward trend, rising monotonically as the training progresses and the policy matures. This consistent increase confirms that MID effectively captures the expansion of the policy's robustness boundary, successfully distinguishing between under-trained and converged models by identifying the precise difficulty threshold where the policy's performance begins to saturate.

## C. Applications

TVS has proven to be a powerful tool for in-depth evaluation of imitation learning policies, owing to its difficulty-aware nature. Nevertheless, beyond assessing model performance across motions of varying difficulty and deriving insightful metrics (e.g., MID and DSJE), TVS holds significant potential for broader applications, including:

1. *Difficulty-guided curriculum learning*: Simpler motions (those with lower TVS scores) are easier to learn and can be used in early training stages, with progressively higher-difficulty actions introduced later to improve learning efficiency and robustness.

2. *Generalization to other humanoid robots.*

3. *Detection of flawed motions in mocap datasets*: Many motion capture (mocap) systems produce noisy or physically implausible data, especially when sourced from video or IMU-based systems (Mu et al., 2025; Zuo et al., 2025; Albanis et al., 2023). These flawed motions can be detected using TVS, as unusually high TVS values may indicate corrupted or unrealistic motions (e.g., unnatural body contortions that violate physical constraints).

Preliminary results of these applications are shown below.

### C.1. Difficulty-Guided Curriculum Learning

Curriculum learning (Bengio et al., 2009) is a training paradigm that begins with simpler examples and progressively introduces more challenging ones, mimicking the natural progression of human learning. This structured, difficulty-ordered presentation of data has been shown to significantly accelerate convergence and enhance generalization in a wide range of tasks (Soviany et al., 2022). However, existing imitation-learning-based humanoid control approaches almost exclusively rely on random sampling of training motions, with virtually no attempts at curriculum learning. We argue that this gap stems from the absence of an explicit, quantitative definition of motion difficulty in prior work.

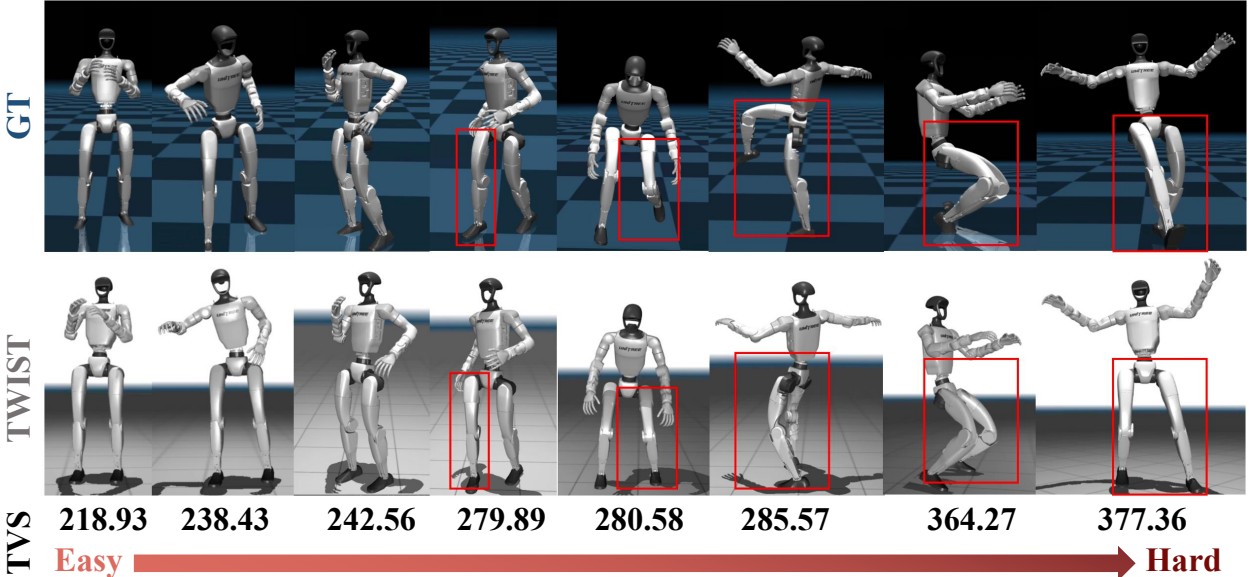

*Figure 8.* Qualitative results of the Unitree G1-based TWIST (Ze et al., 2025) controller on motions of increasing difficulty (easy → hard, left → right) as assessed by TVS. Red boxes highlight joints of noticeable imitation error.

In this section, we present a simple yet revealing curriculum learning experiment. This experiment demonstrates that explicitly training the agent with motions ordered from easy to hard according to TVS leads to superior performance. Specifically, we sample 1000 motion sequences for each of three difficulty ranges from the MD-AMASS dataset: TVS ≤ 200 (easy), 200 ≤ TVS < 300 (medium), and TVS ≥ 300 (hard). We compare two training protocols using the PHC framework with a single primitive policy:

- **Baseline (Tab. 7 w/o CL)**: All 3,000 sequences are mixed and trained jointly for 9,000 episodes.

- **Curriculum Learning (Tab. 7 Ours)**: Training proceeds in three stages of 3,000 episodes each: (1) only easy motions TVS < 200, (2) easy + medium motions TVS < 300, and (3) all motions (full dataset).

We evaluate both models using standard metrics established in PHC: root-relative mean per-joint position error (MPJPE-L), global MPJPE (MPJPE-G), acceleration distance (Acc-Dist), and velocity distance (Vel-Dist). Lower values indicate better performance. As shown in Table 7, the curriculum guided by TVS consistently outperforms the baseline across *all* metrics on the overall test set. Moreover, performance improvements are observed across all difficulty subgroups. We attribute this gain to the difficulty-aware progressive learning schedule enabled by TVS. These results highlight the practical value of our TVS benchmark in guiding the training of humanoid control policies.

### C.2. Generalization to Other Humanoid Robots

In the main experiments, we follow the common practice of UHC (Luo et al., 2021) and PHC (Luo et al., 2023a) by adopting an SMPL-based humanoid model as the controlled robot. However, given the structural similarity among humanoid robots, motions identified as challenging for the SMPL-based morphology are likely to pose comparable difficulties for other humanoid robots, suggesting strong generalization potential of our difficulty metric across different humanoid robots. In this section, we provide a preliminary demonstration of this generalization capability. We retarget motions of increasing difficulty (originally evaluated on the SMPL model) to the Unitree G1 (Unitree Robotics, 2024) robot and qualitatively assess the imitation performance using the state-of-the-art G1-based controller TWIST (Ze et al., 2025). As shown in Fig. 8, on relatively easy motions (left three columns), the policy reproduces the reference motion with minimal error; As TVS increases, noticeable failures emerge, such as incomplete leg rotation during turning (fourth column left), insufficient kick height (third column right), and, in the most challenging dance sequence (first column right), near-complete paralysis of both legs. The results reveal a consistent trend: imitation quality systematically degrades as TVS increases, confirming that the motion difficulty measured by our method on SMPL-structured data reliably generalizes to the G1 robot.

Furthermore, since our method is grounded in rigid-body dynamics, the proposed TVS is, in principle, applicable to other humanoid robotics and even systems beyond humanoid morphologies. We believe that further exploration of our benchmark on a broader range of robotic systems opens substantial and exciting avenues for future research.

