# OpenReview forum: "Distinguishing Imitation Error from Intrinsic Motion Learning Difficulty"
_ICML.cc/2026/Conference — ICML 2026 regular_

### Official Review · Reviewer_YFtL · 2026-02-24

**Soundness:** 3
**Presentation:** 4
**Significance:** 3
**Originality:** 3
**Overall Recommendation:** 4
**Confidence:** 4

**Summary:**

This paper introduces a highly relevant and well-motivated metric for the physics-based humanoid control community. The authors successfully identify a persistent ambiguity in imitation learning evaluations: existing similarity metrics (like MPJPE) conflate a policy's tracking performance with the intrinsic kinematic and dynamic difficulty of the reference motion. By proposing the Torque Variation Score (TVS), the authors provide a physics-grounded, policy-agnostic tool to decouple these two factors. The methodology is theoretically sound, directly linking rigid-body dynamics and torque sensitivity to reinforcement learning optimization challenges such as flat reward landscapes. Overall, this paper offers immediate practical utility for researchers working on robust humanoid control.

**Compliance With Llm Reviewing Policy:**

Affirmed.

**Final Justification:**

Thanks for the detailed rebuttal by the authors, my concerns have been adequately solved. I will keep my positive rating.

**Key Questions For Authors:**

1. In Section 4.2.4, the final Torque Variation Score is computed using a weighted sum of the three diversity terms, with the weights described as "empirically derived". How sensitive is the final TVS ranking to the selection of these specific hyperparameters? Furthermore, when demonstrating generalization to new robot morphologies like the Unitree G1, did these weights require morphology-specific re-
calibration, or were the original SMPL-derived weights used?
2. The theoretical bridge between the proposed rigid-body dynamics metric and actual policy failure relies on the assertion that high-TV motions induce "flat reward landscapes" and a "vanishing gradient" effect. While intuitively sound, this remains a purely theoretical claim in the text. Could you provide empirical measurements, such as tracking gradient variance, gradient norms, or visualizing the reward landscape over localized pose perturbations during the training of UHC or PHC+ to substantiate this optimization collapse?

**Limitations:**

yes

**Strengths And Weaknesses:**

# Strengths
1. The formulation of TVS is elegantly derived from the fundamental equations of rigid-body motion. By aggregating Spectral Diversity (overall torque volume), Variance Diversity (distribution across joints), and Segment Diversity (temporal variability), the metric comprehensively captures the dimensional constraints that make certain motions inherently harder to optimize. The mathematical derivation bridging the Gram determinant to torque space volume is rigorous.
2. The authors provide convincing evidence of TVS's utility by evaluating it across methods like UHC and PHC+ using a massive sample size from the AMASS dataset. The consistent positive correlation between TVS and imitation error strongly validates the metric's core hypothesis.
3. The preliminary results demonstrating TVS's applicability beyond standard SMPL models to real-world humanoid robot kinematics (specifically the Unitree G1 using the TWIST controller) elevate the paper's impact. This proves that the difficulty principles captured by TVS represent fundamental rigid-body constraints rather than artifacts of a specific simulation morphology

# Weakness
1. While TVS accurately characterizes theoretical learning difficulty, it assumes idealized rigid-body dynamics. In physical humanoid deployment, control challenges often stem from hardware-specific constraints, such as actuator torque limits, joint velocity saturation, and bandwidth limitations. Incorporating terms that penalize motions pushing against these physical actuator limits would bridge the gap between simulation difficulty and sim-to-real transfer difficulty.
2. The final TVS calculation relies on a weighted sum of the three diversity metrics, where the weights are described merely as "empirically derived". The manuscript would benefit from a dedicated sensitivity analysis justifying these weight selections. We need to know how robust the metric is to variations in these hyperparameters, or if the weights need to be re-tuned when transferring TVS to a completely different humanoid morphology.

---

> ### Author Rebuttal · Authors · 2026-03-31
>
> We sincerely thank you for the appreciation of our rigorous mathematical derivations, and the recognition of TVS's practical utility for the humanoid control community. We address your questions below.
>
> ## **Weakness 1. Physical Humanoid Deployment and Sim-to-Real Transfer Difficulty**
> We appreciate the insight into hardware-specific constraints but respectfully emphasize that establishing a clear theoretical foundation is a necessary first step before incorporating such factors. The progression from idealized theoretical analysis and simulation to practical deployment is standard in scientific research and helps isolate the core properties of the metric.
> For example, in robotics and physics-based control, seminal imitation learning works such as DeepMimic (Peng et al., 2018) first establish algorithmic viability in simulation. Subsequent sim-to-real efforts (e.g., Peng et al., 2020) then extend these frameworks by incorporating actuator dynamics and hardware constraints for physical deployment.
> Therefore, we believe our work provides the humanoid control community with a solid, morphology-agnostic theoretical foundation and establishes a clear, rigorous pathway for future engineering refinements, serving as a concrete initial step in this direction. We will include a discussion of hardware-specific constraints as part of future work in the revision.
>
> ## **Weakness 2 & Key Question 1. Hyperparameter Sensitivity and Cross-Morphology Generalization**
>
>   Following the ablation protocol established in the main manuscript, we systematically evaluated the sensitivity of the TVS hyperparameters ($w_1, w_2, w_3$). Specifically, we measured the correlation coefficients (Pearson, Spearman, and Kendall) between the resulting TVS and the imitation error (MPJPE) using the UHC policy.
>   As shown in the table below, we assigned a larger, dominant weight to one specific component while keeping the other two components smaller and equal (e.g., 0.5, 0.25, 0.25), allowing us to observe if over-relying on any single term degrades the metric's overall correlation.
>
>  | $w_1$ | $w_2$ | $w_3$ | Pearson | Spearman | Kendall |
> | :---: | :---: | :---: | :---: | :---: | :---: |
> | 0.4 | 0.3 | 0.3 | 0.4823 | **0.6586** | **0.4747** |
> | 0.3 | 0.4 | 0.3 | 0.4832 | 0.6502 | 0.4740 |
> | 0.3 | 0.4 | 0.4 | **0.4838** | 0.6584 | 0.4745 |
> | 0.5 | 0.25 | 0.25 | 0.4760 | 0.6550 | 0.4716 |
> | 0.25 | 0.5 | 0.25 | 0.4209 | 0.6233 | 0.4369 |
> | 0.25 | 0.25 | 0.5 | 0.4781 | 0.6562 | 0.4725 |
> | 0.7 | 0.15 | 0.15 | 0.4657 | 0.6444 | 0.4628 |
> | 0.15 | 0.7 | 0.15 | 0.4473 | 0.6172 | 0.4409 |
> | 0.15 | 0.15 | 0.7 | 0.4616 | 0.6478 | 0.4653 |
>
>   We chose $w_1, w_2, w_3 = [0.4, 0.3, 0.3]$ in our final implementations.
>   Note that even when the weights are subjected to extreme skewness (e.g., up to 0.7 for a single component), the correlation coefficients remain stable (e.g., the Spearman correlation consistently stays above 0.6). This empirical evidence substantiates that our proposed TVS metric is highly robust to hyperparameter selections.
>   Thanks to this hyperparameter-robustness, in our Appendix C.2 (Generalization to Other Humanoid Robots), we did not perform any morphology-specific recalibration. The same weights derived from the SMPL model were applied directly to the G1 kinematics. The fact that TVS generalizes to a different humanoid structure without re-tuning further indicates that it captures underlying rigid-body constraints rather than overfitting to a specific morphology, which further demonstrates it effectiveness.
>
> ## **Key Question 2. Empirical Evidence for "Flat Reward Landscapes"**
> To measure this optimization collapse, we tracked the reward output of the pre-trained PHC+ policy under progressively increasing random pose perturbations (magnitude: Perturbation  1 < 2 < 3) for different motion categories as follows:
>
> *Low TVS (TVS < 150):*
>
> | | Reward | Reward Drop (relative to above) |
> | :--- | :---: | :---: |
> | No Perturbation | 263.99 | - |
> | Perturbation 1 | 253.70 | -10.29 |
> | Perturbation 2 | 247.46 | -6.24 |
> | Perturbation 3 | 224.75 | -22.71 |
>
> *High TVS (TVS > 350):*
>
> | | Reward | Reward Drop (relative to above) |
> | :--- | :---: | :---: |
> | No Perturbation | 190.63 | - |
> | Perturbation 1 | 187.63 | -3.00 |
> | Perturbation 2 | 187.23 | -0.40 |
> | Perturbation 3 | 187.74 | +0.51 |
>
> It can be observed that:
> * For Low-TVS motions, the reward drops significantly and monotonically as the perturbation increases. This provides a clear, informative signal for the policy to learn the correct reference pose.
> * In contrast, for High-TVS motions, the reward remains nearly flat (hovering around 187) regardless of the perturbation increase. In this case, the policy explores drastically different perturbed states but receives virtually indistinguishable rewards, making the learning significantly more challenging.
> This empirical evidence validates the "flat reward landscape" claim in our paper. We will include a more detailed analysis in the revision.

---

> > ### Author Rebuttal · Reviewer_YFtL · 2026-04-02
> >
> > Thanks for the detailed rebuttal by the authors, my concerns have been adequately solved.

---

### Official Review · Reviewer_mp4u · 2026-03-02

**Soundness:** 3
**Presentation:** 3
**Significance:** 3
**Originality:** 3
**Overall Recommendation:** 5
**Confidence:** 4

**Summary:**

The authors proposed a Torque Variation Score for evaluating if the error in physics based imitation is introduced by task difficulty or policy limitation. Specifically, in the case where correcting the error requires higher torque (or higher variation in torque), the error is more related to task difficulty. Otherwise, it is due to policy limitation.

**Compliance With Llm Reviewing Policy:**

Affirmed.

**Final Justification:**

In the rebuttal, the authors have addressed my concerns properly. After considering the responses from the authors and the reviews from the other reviewers, I decide to raise my score to accept.

**Key Questions For Authors:**

Please address the weaknesses part of my review.

**Limitations:**

The authors did not include a limitation section, I would recommend them to include one, which will be helpful for future readers.

**Strengths And Weaknesses:**

Strengths:

1. The Torque Variation Score proposed by the authors can be a valuable tool for the research community.

2. The writing and presentation of the paper is good.


Weaknesses:

I think to properly validate the claim (in the case where correcting the error requires higher torque, the error is more related to task difficulty. Otherwise, it is due to policy limitation), it is important to include two additional experiments: (1) explicitly use some weaker policy networks, for example policy with reduced parameter count and validate that the error increases without increase in TVS. (2) Introduce more training data which requires large torque correction and demonstrate that both TVS and imitation error increases.

---

> ### Author Rebuttal · Authors · 2026-03-31
>
> We sincerely thank you for recognizing the value of the Torque Variation Score (TVS) to the research community. We provide the required additional experiments below.
>
> ## **Validation with Weaker Policies (Requested Experiment 1)**
>
>   We leverage early training checkpoints of PHC+, which naturally serve as inherently weaker, under-trained policy networks. Testing on motion samples with constant TVS, we report the imitation error MPJPE across these progressively stronger policies as follows:
>
> | Motion Sample | Epoch 1500 | Epoch 3000 | Epoch 4500 | Epoch 6000 | Epoch 7500 | Epoch 9000 |
> | :--- | :---: | :---: | :---: | :---: | :---: | :---: |
> | Sample 1 (TVS = 324) | 122.38 | 100.56 | 98.99 | 90.14 | 89.27 | 70.37 |
> | Sample 2 (TVS = 224) | 110.74 | 97.38 | 91.08 | 89.65 | 72.91 | 67.95 |
> | Sample 3 (TVS = 172) | 43.59 | 39.16 | 38.91 | 38.65 | 36.75 | 35.30 |
>
> It can be observed that for weaker policies (i.e., from earlier training epochs), the evaluated MPJPE on motions with identical TVS is consistently lower, which validates our claims.
>
> ## **Validation with Additional Data Requiring Large Torque Correction (Requested Experiment 2)**
>
>   To empirically validate that motions requiring large torque corrections yield both high TVS and elevated imitation errors, we introduce additional test samples from sparse-IMU-based MoCap data. Note that these specific sequences contain observable physical artifacts, such as unnatural joint bending, to meet the requirement for large torque corrections. We evaluated the PHC+ policy on these extreme sequences, and the quantitative results are as follows:
>
> | Motion | TVS | MPJPE |
> | :--- | :---: | :---: |
> | Leg stretch | 327.19 | 100.89 |
> | Lean forward | 335.00 | 145.55 |
> | Stamp | 337.65 | 150.73 |
> | Run | 345.81 | 161.09 |
> | Jump | 354.25 | 178.92 |
> | Run backward | 360.00 | 189.63 |
> | Bend | 361.46 | 246.88 |
>
> It can be observed that both TVS and imitation error increases, which again validates our claims.

---

> > ### Author Rebuttal · Reviewer_mp4u · 2026-04-03
> >
> > After considering the responses from the authors and the reviews from the other reviewers, I decide to raise my score to accept.

---

### Official Review · Reviewer_8Qwi · 2026-03-12

**Soundness:** 3
**Presentation:** 3
**Significance:** 3
**Originality:** 3
**Overall Recommendation:** 5
**Confidence:** 4

**Summary:**

This work proposes the Torque Variation Score (TVS), a physics-grounded metric to quantify the intrinsic difficulty of humanoid motions independently of a learned policy performance. By analyzing the sensitivity of required torques to minor pose perturbations, TVS enables a principled error metric that heavily correlates with motion difficulty. Using TVS, one can filter better quantify the inherent difficulty of motion for policy learning, as well as filtering out bad motion. Analysis on two motion trackers shows that TVS correlates well with motion difficulty and policy learning variation and can be a powerful offline tool for dataset analysis and motion learning prioritization.

**Compliance With Llm Reviewing Policy:**

Affirmed.

**Final Justification:**

I think the authors for the detailed reply. My concerns are addressed and I will keep my score.

**Key Questions For Authors:**

In Sec.4.2.2 L256, should it be smaller variance significes lower motion difficulty?

**Limitations:**

Yes.

**Strengths And Weaknesses:**

## Strength

- This work attempts to address an important question in humanoid motion tracking: distinguishing between policy-induced errors and motion-inherent tracking difficulties. Current popular metrics such as MPJPE only show policy performance, but do not provide the reason behind such performance. TVS provides a dynamics-grounded way to explain why motion tracking failed based on the underlying dynamics.
- The metric, TVS, derived from rigid-body dynamics, provides a solid mathematical derivation for why certain motions are inherently hard to learn and unstable.
- The application of TVS to flawed motion detection is practical, and allows cleaning up motion capture datasets as well as restarting motion datasets.
- The diagnostic tools introduced, such as DSJE and MID, can be future tools to better quantify the motion tracker’s performance.

## Weakness

- UHC utilizes residual-force control, which leads to the comparison not being entirely fair. Also, the simulator used is different (mujoco vs isaacgym).
- By relying on an analytical inverse-dynamics formulation to compute torque sensitivity, TVS does not properly model the contact with the ground. It is unclear if the metric is reliable for motions where the main challenge is modeling contact forces.

---

> ### Author Rebuttal · Authors · 2026-03-31
>
> We sincerely thank you for your positive evaluation, and the recognition of TVS's practical applications in dataset cleanup and policy diagnosis.
>
> ## **Weakness 1. Fairness of Evaluation Across Policies**
> We appreciate the careful observation. We would like to clarify that our evaluation across different frameworks is fair and scientifically valid. The perceived unfairness stems primarily from a terminology ambiguity ("policy" vs. "method"), and the software differences between simulators do not alter the fundamental physical laws upon which TVS is computed. We detail these points below.
> * **Residual forces.** We believe the confusion arises from our use of the term "policy" rather than "method". In our context, residual-force control is an integral, inseparable part of the UHC framework, removing it would fundamentally cripple the policy, and we have to evaluate the entire method to accurately reflect its true imitation error. We will rephrase this in our revision.
> * **Different Simulators.** Although MuJoCo and Isaac Gym differ in their implementation details (e.g., numerical solvers), they are grounded in the same rigid-body dynamics and control principles. In both cases, policy actions are translated into joint torques via Proportional-Derivative (PD) control under consistent physical assumptions such as mass, inertia, and gravity. As a result, the formulation of TVS remains well-defined across simulators. Moreover, the strong correlation between TVS and imitation error observed in both simulators suggests that the metric reflects general physical constraints rather than simulator-specific artifacts.
> ## **Weakness 2. Contact with the Ground.**
> Good point! We intentionally omit ground contact forces, as they are mathematically negligible for the daily human motions we study. Specifically, from the rigid-body dynamics equation, the mapping $F$ from state to torque is defined as:
>
> $$F(q, \dot{q}, \ddot{q}) = M(q)\ddot{q} + h(q, \dot{q}) - f_{ext},$$
>
> when computing the torque sensitivity via the Jacobian matrix $J = dF$, the formulation requires computing the derivatives of the internal dynamics terms $(dM,dh)$; and the external contact force $(df_{ext})$.
> According to (Winter, 2009), external ground reaction forces $f_{ext}$ during daily motions are predominantly governed by Center of Mass (CoM) kinematics and gravity. Because small local pose perturbations do not significantly alter the global CoM trajectory, the variation in external contact forces $df_{ext}$ is infinitesimally small and mathematically negligible compared to $dM$ and $dh$ .
>
> To provide an intuitive quantitative comparison, we validate this using the real-world dataset GroundLink [Han et al., SIGGRAPH Asia 2023]. During a walking motion, the peak variation of external forces ($|\frac{\partial f_{ext}}{\partial q}|$) is merely 2.4. Even for highly dynamic motions with larger impact frequency, such as hopping, this peak only reaches 7.9. Conversely, on a regular walking motion, $|\frac{\partial (M\ddot{q})}{\partial q}|$ averages 1046, and $|\frac{\partial h}{\partial q}|$ peaks at 1288. This extreme ratio (~0.23%) demonstrates that $df_{ext}$ is mathematically safe to ignore.
>
> While one might hypothesize extreme theoretical scenarios where massive, discontinuous contact forces could influence the computation, our observations suggest such cases are likely non-existent in natural human biomechanics.
>
> ## **Key Question. Clarification on Variance Diversity**
> Good point again! While seemingly counter-intuitive, this observation in fact provides strong supporting evidence for our core contribution. Specifically, by "smaller variance signifies higher motion difficulty" (Sec. 4.2.2 L256), we mean that Variance Diversity measures the disparity or spread of torque variations across different joints.
> * If the variance is large, it means the high torque sensitivity is concentrated on only one or a few specific joints (e.g., just the waving arm), while the rest of the body remains stable. This is relatively easy for an RL policy to learn.
> * Conversely, if the variance is small, it implies that the torque variations are uniformly distributed across all joints. This means any minor pose perturbation requires the policy to make simultaneous, delicate torque adjustments across the entire body (e.g., a precarious full-body balancing act). Coordinating the whole body simultaneously is intrinsically much harder for the policy than adjusting a single limb.
>
> Therefore, a smaller variance across joints indicates a higher holistic instability, which directly translates to a higher motion learning difficulty. We will further clarify this intuition in the revision to improve clarity and avoid potential misunderstandings.

---

> > ### Author Rebuttal · Reviewer_8Qwi · 2026-04-06
> >
> > I think the authors for the detailed reply. My concerns are addressed, and I will keep my score.

---

### Official Review · Reviewer_uoNR · 2026-03-13

**Soundness:** 3
**Presentation:** 2
**Significance:** 2
**Originality:** 2
**Overall Recommendation:** 5
**Confidence:** 3

**Summary:**

This paper proposes Torque Variation Score (TVS), a torque-based metric to measure the difficulty of motion learning. Based on the rigid body dynamics, the authors compute the torque variations by perturbing the motion sequences. They use torque variations to introduce 3 metrics for difficulty measurement: spectral diversity to capture the effective dimensionality of the feasible torque space, variance diversity to capture the joint-wise torque variation, and segment diversity to capture the temporal difficulty. TVS is the weighted sum of these metrics.  Experiment results demonstrate that the TVS score correlates the imitation error from commonly use imitation learning polices. TVS can also be used for other applications such as flawed motion detection and difficulty-guided curriculum learning.

**Compliance With Llm Reviewing Policy:**

Affirmed.

**Final Justification:**

This paper provide a reasonable metric to quatify the motion imitation diffculty, with sufficient emprical evaluation as well as demonstrations of down-stream tasks. My concerns are addressed during rebuttal.

**Key Questions For Authors:**

1. From Figure 3 seems that some motions with similar imitation errors have a large discrepancy in TVS, such as motions with ~50 mm MPJPE-G. Can you report what these motions are and explain the large score gap?

2. Does the TVS score sensitive to the record framerate of the motion?

**Limitations:**

The authors adequately discussed the limitations and potential negative societal impact

**Strengths And Weaknesses:**

**Strength**

1. This paper proposes a reasonable metric to qualify the motion difficulty, which helps to identify the key factors of imitation error.

2. The paper is clear and easy to follow.

3. The experimental results demonstrate correlation between TVS and imitation error, and the applications of flawed motion detection and difficulty-guided curriculum learning shows the practicality of TVS in imitation learning pipeline.

**Weakness**

1. The proposed TVS has several hyperparameters such as K in segment diversity and the score weights. The authors did not report the used hyperparameter settings, and did not perform ablation study over these hyperparameters.

2. The correlation of TVS and imitation error seems sensitive to the used policy, where the correlation score has large gap between UHC and PHC+.

---

> ### Author Rebuttal · Authors · 2026-03-31
>
> Thank you for recognizing the value of TVS and for the constructive feedback. Below we address your questions in detail.
> ## **Weakness 1. Ablation Study on the Choice of Hyperparameters.**
> Following the ablation protocol in the main manuscript, we evaluated the choice of hyperparameters $w_1, w_2, w_3$ and $K$. Below, we measured the correlation coefficients (Pearson, Spearman, and Kendall) between the TVS and the imitation error (MPJPE) using UHC.
> * $w_1, w_2, w_3$. We assign a larger, dominant weight to one specific component while keeping the other two components smaller and equal (e.g., 0.5, 0.25, 0.25), observing if over-relying on any single factor degrades the metric's overall correlation.
>
> | $w_1$ | $w_2$ | $w_3$ | Pearson | Spearman | Kendall |
> | :---: | :---: | :---: | :---: | :---: | :---: |
> | 0.4 | 0.3 | 0.3 | 0.4823 | **0.6586** | **0.4747** |
> | 0.3 | 0.4 | 0.3 | 0.4832 | 0.6502 | 0.4740 |
> | 0.3 | 0.4 | 0.4 | **0.4838** | 0.6584 | 0.4745 |
> | 0.5 | 0.25 | 0.25 | 0.4760 | 0.6550 | 0.4716 |
> | 0.25 | 0.5 | 0.25 | 0.4209 | 0.6233 | 0.4369 |
> | 0.25 | 0.25 | 0.5 | 0.4781 | 0.6562 | 0.4725 |
> | 0.7 | 0.15 | 0.15 | 0.4657 | 0.6444 | 0.4628 |
> | 0.15 | 0.7 | 0.15 | 0.4473 | 0.6172 | 0.4409 |
> | 0.15 | 0.15 | 0.7 | 0.4616 | 0.6478 | 0.4653 |
>
> We chose $w_1, w_2, w_3$ = [0.4, 0.3, 0.3] in our implementations.
> Note that even when the weights are subjected to extreme skewness (e.g., up to **0.7** for a single component), the correlation coefficients remain stable (e.g., the Spearman correlation consistently stays over **0.6**). This empirical evidence substantiates that our proposed TVS metric is robust to hyperparameter selections.
> * **$K$ in Segment Diversity:** Furthermore, we applied the same evaluation logic to test the sensitivity of the hyperparameter $K$ used in Segment Diversity.
>
> | $K$ | Pearson | Spearman | Kendall |
> | :---: | :---: | :---: | :---: |
> | 2 | 0.4618 | 0.6524 | 0.4552 |
> | 3 | 0.4691 | 0.6534 | 0.4572 |
> | 4 | **0.4823** | **0.6586** | **0.4747** |
> | 5 | 0.4763 | 0.6534 | 0.4542 |
> | 6 | 0.4793 | 0.6586 | 0.4501 |
> | 10 | 0.4813 | 0.6554 | 0.4491 |
> | 15 | 0.4856 | 0.6516 | 0.4470 |
>
> We use $K=4$ in our implementations. The results indicate that our method is also largely insensitive to the choice of $K$.
> ## **Weakness 2. Sensitivity to Used Policy / Gap in Correlation**
> We hope to clarify that correlation strength is typically interpreted using standardized effect size conventions rather than by direct comparison of raw values. For example, according to the established statistical guidelines by *Jacob Cohen in Statistical Power Analysis for the Behavioral Sciences*, a Spearman correlation of r = 0.10 is considered "Small", r = 0.30 is "Medium", and r ≥ 0.50 is "Large/Strong". In our experiments, the Spearman correlations for both policies consistently exceed 0.60 (0.658 for UHC and 0.820 for PHC+), demonstrating a robust and strong correlation across the board. In other words, the observed "strong correlation" between TVS and imitation errors is insensitive to the specific policy used.
> ## **Key Question 1. Similar Errors vs. Large TVS Discrepancy**
> Good point! This insightful observation highlights an inherent limitation of standard geometric metrics like MPJPE, which further demonstrates the strengths of our TVS. That is, while MPJPE remains the reliable and widely accepted standard for capturing overall accuracy, its nature as an unweighted average means it can sometimes mathematically obscure localized physical failures.
>  Specifically, visual inspection of specific motions at the ~50 mm MPJPE mark reveals distinct failure modes:
> * Low TVS (~50 mm MPJPE, e.g., walking): The error is typically a uniform global joint drift. The policy successfully maintains balance, and the motion remains dynamically stable and visually plausible.
> * High TVS (~50 mm MPJPE, e.g., high kicks): The error often stems from a severe, localized dynamic collapse. For instance, while attempting a high kick, the policy fails to generate the extreme required torque, causing the supporting knee to bend unnaturally by 200+ mm. However, when averaged across all 24 joints, this local failure is mathematically diluted to ~50 mm.
>   These results demonstrate that TVS can capture failure modes that are not reflected in MPJPE, further highlighting its complementary strengths. We will include more discussions in the revision.
>
> ## **Key Question 2. Sensitivity to Framerate**
>
> TVS is not sensitive to the recording framerate of the motion. In our data processing pipeline, we align and resample all input motion sequences to a unified 60fps framerate before computing torque variations. As shown in the table below, resampling the same motion samples recorded at different framerates to 60fps yields nearly identical TVS scores.
>
> | Motion Sample | 30fps | 60fps (default) | 120fps |
> | :--- | :---: | :---: | :---: |
> | hand_waving | 188.58 | 188.58 | 188.58 |
> | tennis | 273.48 | 273.48 | 273.48 |
> | jump | 333.87 | 333.87 | 333.87 |

---

> > ### Author Rebuttal · Reviewer_uoNR · 2026-04-02
> >
> > I thank the authors for their response and sensitive analysis. My concerns are addressed. I have raised my score to 5 (accept).

---

### Decision · Program_Chairs · 2026-04-30

**Decision:**

Accept (regular)

**Comment:**

This paper presents Torque Variation Score (TVS), a physics-grounded metric quantifying intrinsic motion learning difficulty independent of policy performance, enabling principled error attribution in humanoid imitation learning. All reviewers converge on the contribution's elegance and utility: strong Spearman correlations across policies (UHC, PHC+), practical applications (flawed motion detection, curriculum learning), and generalization to Unitree G1. The rebuttal was exceptionally effective—three reviewers raised scores from 4 to 5 after receiving sensitivity analyses, framerate robustness tests, and contact-force quantification (0.23% contribution). Final scores (5,5,5,4) reflect strong consensus. A clean, well-scoped contribution that provides a needed diagnostic tool for the community.